The effect of cumulative ecological risk on migrant children’s Internet game addiction: a moderated mediation model

Lin Zhengzheng linzhengzheng@zjnu.edu.cn 1 2
Zha Ying zhayingzjnu@163.com 3
1 School of Psychology, Zhejiang Normal University , Jinhua , Zhejiang , China
2 School of Normal Education, Longyan University , Longyan , Fujian , China
3 Tin Ka Ping Moral Education Research Center, Zhejiang Normal University , Jinhua , Zhejiang , China
Myers Catherine
Electronic publication date: 2025 Aug 28
Publication date: 2025
Volume: 13
Electronic Location ID: e19787
Received 2024 Oct 8; Accepted 2025 Jul 1
Copyright: ©2025 Lin and Zha
Copyright year: 2025
Copyright holder: Lin and Zha
License: This is an open access article distributed under the terms of the Creative Commons Attribution License, which permits unrestricted use, distribution, reproduction and adaptation in any medium and for any purpose provided that it is properly attributed. For attribution, the original author(s), title, publication source (PeerJ) and either DOI or URL of the article must be cited.
License URL: https://creativecommons.org/licenses/by/4.0/

Keywords: Cumulative ecological risk, Internet game addiction, Self-educational expectations, Gender, Migrant children

Funding: The project of 14th Five-Year Plan of Educational Science in Fujian Province, China FJJKBK23-117 FJJKBK24-037 The Fujian Social Science Foundation Project FJ2025C179 This study were supported by the project of 14th Five-Year Plan of Educational Science in Fujian Province, China (FJJKBK23-117; FJJKBK24-037), and the Fujian Social Science Foundation Project (FJ2025C179). The funders had no role in study design, data collection and analysis, decision to publish, or preparation of the manuscript.

==============================
Background

Internet games are becoming a popular form of entertainment. However, overindulgence in online games may lead to online game addiction, which may have a negative impact on the health of adolescents. As a special group of adolescents, migrant children’s Internet game addiction has also attracted much attention. This study aimed to explore the effects of cumulative ecological risk on migrant children’s Internet game addiction, and to investigate the mediating role of self-educational expectations and the moderating role of gender.

Methods

A questionnaire survey method was employed in this study. Using convenience sampling, a total of 314 migrant children completed the cumulative ecological risks scale, self-education expectations scale, and Internet game addiction scale.

Results

The findings showed that cumulative ecological risk had a significant effect on migrant children’s Internet game addiction, and self-education expectations partially mediating the relationship between cumulative ecological risk and Internet game addiction. The latter part of the mediating effect in the pathway “cumulative ecological risk → self-educational expectations → Internet game addiction” is moderated by gender. Specifically, the impact of self-educational expectations on game addiction is stronger for male students compared to female students.

Conclusion

Comprehensive measures should be taken to address the problem of online game addiction among migrant children, including reducing the ecological risk faced by migrant children, raising self-education expectations and paying attention to gender differences.

Introduction

With the continuous development of the Internet, online games have become one of the most sought-after recreational activities among adolescents. Although parents, schools, and other societal authorities monitor the Internet use of adolescents, they are still at a high risk of becoming addicted to Internet games (Mihara & Higuchi, 2017). According to the 52nd Statistical Report on Internet Development in China, as of June 2023, the number of online game users in China reached 550 million, accounting for 51% of the total number of Internet users, the proportion of Internet users aged 10–19 years in China reached 13.9% (China Internet Network Information Center, 2023). As an addictive activity on the Internet, online gaming has a particularly negative impact on individual development in terms of psychophysiological functioning (Kuss, 2013); for example, insomnia, low academic performance, depression, loneliness, negative emotional experiences, and even the risk of suicide (Chen et al., 2020; Gundogdu & Eroglu, 2022).

In 2020, there were 53.19 million migrant children aged 0–14 in China. There is one migrant child for every four children (Han, 2023). Many of the risk factors that migrant children face when they enter the city can be harmful to their behavioral development. For instance, migrant families generally have low income and the parents have low levels of education and face life pressures, family collaboration and communication difficulties, and a lack of a supportive environment for their children to grow up in. As a result of their mobility, they may suffer from unfair treatment by teachers and discrimination by peers at school, and become alienated from their peers and teachers. The proportion of problematic behaviors is higher among migrant children than among those in the general population (Belhadj, Koglin & Petermann, 2014).

Therefore, this study aims to investigate the environmental factors (cumulative ecological risk) and individual factors (self-education expectation, gender) affecting migrant children’s addiction to online games as well as their psychological mechanisms. This holds high reference value for exploring preventive measures against online game addiction among migrant children.

Cumulative ecological risk

According to ecological systems theory, individual development is nested within a series of interacting environmental systems. Families, schools, and peers are important ecological subsystems that influence the healthy development of young people (Damon & Lerner, 1998). Numerous studies have also confirmed the influence of family, school, and peers on adolescents’ addictive behaviors.

On the family side, with inadequate parental support, adolescents may turn to the Internet for psychological solace (Karaer & Akdemir, 2019). Mothers with higher levels of education have more effective strategies to regulate their children’s use of digital games (Karaca, Aral & Kaya, 2025). Toker & Baturay (2016) pointed out that family socioeconomic status affect Internet game addiction significantly. Parental marital conflict can lead to social maladjustment and problematic behaviors in individuals. Wang et al. (2014) showed that perceived family disharmony significantly associated with adolescent game addiction.

In terms of schools and peers, poor teacher–student and peer relationships can also adversely affect adolescents’ Internet use behavior (Badenes-Ribera et al., 2019; Peng et al., 2020). Teachers who are indifferent to their pupils are not conducive to matching the needs of migrant children with their environment and may have a negative impact. Migrant children experience higher levels of peer discrimination than local children (Giuliani, Tagliahue & Regalia, 2018). School connectedness is an emotion that allows students feel a sense of belonging and identity formed by mutual attachment with people in the school and actively engage in learning activities (Mulla, Bogen & Orchowski, 2020). Zhu et al. (2015) found that school connectedness and deviant peer affiliation had an impact on adolescents’ gaming addiction after a 1-year follow-up study of 7th graders.

It is evident that ecological risk factors may be one of the important variables influencing the behavioral development of mobile junior high school students. Previous studies have focused on the effect of a single or a few ecological risk factors on adolescent addictive behavior. However, each risk factor does not exist independently, and individuals face multiple risks simultaneously (Giovanelli et al., 2020).

The cumulative risk theory states that adolescents do not face a single set of risk factors, but rather face multiple risk factors from multiple domains that lead to adolescent maldevelopment. The presence of multiple risk factors, as opposed to a single risk factor, may lead to an increased incidence of problematic behaviors in adolescents (Evans, Li & Whipple, 2013). Based on this, the study proposes hypothesis 1: Cumulative ecological risk positively predicts Internet game addiction in migrant children.

Self-educational expectations

The individual-environment interaction theory suggests that negative external environment and intrinsic factors within the individual interact to influence behavior (Thomas, Segal & Hersen, 2006). Self-educational expectations are beliefs about the level of education to be attained in the future, based on one’s actual situation and the learning environment in which one finds oneself as well as one’s long-term academic goals. Adolescence is a critical period for the establishment of self-identity that gradually shapes self-educational expectations for the future. Self-educational expectations are closely related to academic performance and future development (Andrew & Hauser, 2011).

Self-education expectations are positive future-oriented beliefs. Self-system theory suggests that the presence of risk factors in an ecosystem can adversely affect the positive beliefs of an individual’s self-system (Prelow, Weaver & Swenson, 2006). Family is an important factor in influencing children’s self-educational expectations, and some researchers have noted that students from different family economic backgrounds have different self-expectations (Goyette & Xie, 1999). A study found that the less educated the mothers were, the lower their children’s academic goals were (Augustine, 2017). A parent–child relationship with positive emotional responses and feedback allows for the sharing of timely educational information that helps adolescents construct their self-beliefs and expectations about their future. On the contrary, if supportive resources are scarce, migrant children may lose motivation to pursue their growth goals (Hasanah, Susanti & Panjaitan, 2019).

For adolescent students who have been in a school environment for a long time, student-school connectedness promotes their expectations of success. Adolescents with low levels of school connectedness are prone to anxiety, depression and experience insecurity. At the same time, teacher–student relationships are effective in predicting students’ educational expectations (Haimovitz & Henderlong Corpus, 2011). Adolescents with high self-educational expectations devote more time to their studies, more proactively reduce the distractions of extraneous events, and are more likely to have a long-term learning development plan than those with low expectations (Carroll et al., 2009). The expectancy effect states that individuals who receive positive attention dramatically increase their individual potential to achieve desired behavioral goals by creating self-efficacy and high expectations, in turn changing their attributional styles and motivations (Rosenthal & Jacobson, 1968). Self-expectation in academics has a protective effect against risky behavior. High self-educational expectations improve adolescents’ cognitive abilities and social behaviors and promote positive development whereas low educational expectations may result in negative behavioral outcomes. A longitudinal study found that eighth-grade students who set up future academic orientations reduced their drinking behavior in the ninth grade (Lee et al., 2015).

Future time perspective, as an individual’s expectation of future goals, is congruent with self-educational expectations. Future time perspective can predict unhealthy behaviors such as smoking, drinking, and substance abuse (Chittaro & Vianello, 2013; McKay, Percy & Cole, 2013). Cumulative ecological risk increases unhealthy behaviors and problematic behaviors in rural adolescents by decreasing future time insight (Jin, Ji & Li, 2022). Individuals who experience chronic frustration are likely to lack self-control and display oppositional defiance, and these aspects promote negative cycles of further frustration, maladaptation, and ill-being (Vansteenkiste & Ryan, 2013). Thus, we propose hypothesis 2 as follows: Cumulative ecological risk affects migrant children’s Internet gaming addiction through self-education expectations.

Moderating role of gender

There are gender differences in the psychological mechanisms underlying problematic Internet use among adolescents. Impulse control is weaker among males than among females, and the former are more prone to negative risk-taking behavior. It has been reported that males are at a higher risk of Internet game addiction than females (Toker & Baturay, 2016; Dong, Li & Hu, 2023).

Further, gender is an important moderating variable in Internet addiction research (Busch & McCarthy, 2021). For example, gender moderates the relationship between impulsivity and problematic online gaming, with the effect being significant only among males (Su et al., 2019). Self-esteem significantly predicted online gaming addiction among boys and not among girls (Ko et al., 2005). Zhang et al. (2023) found that anxiety had a stronger impact on internet game addiction than girls. Therefore, the role of gender differences in Internet game addiction among migrant children cannot be ignored. Research is needed to develop targeted interventions based on the characteristics of different groups.

When adolescents begin to think seriously about the future, significant differences arise in the levels of their expectations about their future education, but the findings of previous studies have not been consistent (Organization for Economic Co-operation Development, 2019). Wei & Ma (2017) noted that self-education expectations are generally higher for female students than for male students, both in rural and urban areas. At the junior high school level, self-education expectations are more positive among girls than among boys. However, Sulimani-Aidan, Sivan & Davidson-Arad (2017) found that males have higher levels of hope for the future than females. Accordingly, we propose hypothesis 3 as follows: Gender plays a moderating role in the second half of the mediation pathway between self-educational expectations and Internet gaming addiction.

Present study

Adolescents’ online game addiction is closely related to their individual ecological background. However, to our best knowledge, existing studies have focused on the role of a single ecological risk on adolescents’ online addiction, lacking a comprehensive assessment of multiple ecological risks (family, school, etc.). Furthermore, previous studies has largely focused on general groups of children, while studies targeting migrant children are fewer, neglecting the particularities of migrant children in terms of family, school, and social integration. Based on this, the study selected typical and representative ecological risk factors to examine the effect of cumulative ecological risk factors on online game addiction to obtain some meaningful findings. Meanwhile, based on the proposed hypothesis, this study constructed a mediated moderating effect model of cumulative ecological risk leading to Internet game addiction among migrant adolescents, as shown in Fig. 1.

Material and Methods

Participants

Convenience sampling was used to select migrant students in grades 7–9 from five secondary schools in Fujian Province, China. A total of 700 questionnaires were distributed and 692 were collected, with a recovery rate of 98.86%. Nine questionnaires with missing answers and random responses were excluded, resulting in 683 valid questionnaires (validity rate of 98.7%). Refer to the definition by Lin et al. (2009), in this study, “migrant children” refers to students in grades 7–9 who are part of the internally mobile population in China, have moved with their guardians from rural to urban areas for schooling for more than six months, and do not hold urban household registration (hukou). There were 314 migrant children, of which 171 were males and 143 were females; 110 were in grade 7, 101 were in grade 8, and 103 were in grade 9, ages from 12–15 years (M = 13.08, SD = 0.89).

Measures

Demographic information

Basic information, including gender, age, grade level, household location, and time spent living and studying in the city, was collected and analyzed.

Figure 1 Hypothetical model.

Cumulative ecological risk

On the basis of previous studies, representative risk factors in the proximal ecological subsystems of family, school, and peers, which may closely related to online game addiction among migrant children, were selected to construct a cumulative ecological risk index (Li et al., 2016). As shown in Table 1, the 25th or 75th percentile of the scores of each risk variable was used as the critical value, with risk coded as 1 and no risk coded as 0; then, all the scores of the risk factors were summed up to obtain the cumulative ecological risk index (Wade et al., 2015). The measurement tools for each ecological risk factor are described below:

Table 1 Ecological risk description and definition of cumulative ecological risk.

Risk index	Risk areas	Rating	Risk-definition criteria	
FES	Family	1 (totally disagree) to 5 (totally agree)	Above 75th percentile was code as 1 (at risk) and the rest were coded as 0 (no risk).	
FLE	Family	1 (no schooling), 2 (primary schools), 3 (junior high school), 4 (secondary/secondary/vocational), 5 (specialized or undergraduate), 6 (graduate school and above)	Father with educational attainment below upper secondary/secondary/vocational” was coded as 1 (at risk) and the rest were coded as 0 (no risk).	
MLE	Family	1 (no schooling), 2 (primary schools), 3 (junior high school), 4 (secondary/secondary/vocational), 5 (specialized or undergraduate), 6 (graduate school and above)	Mother with educational attainment below upper secondary/secondary/vocational” was coded as 1 (at risk) and the rest were coded as 0 (no risk).	
PMR	Family	1 (totally disagree) to 5 (totally agree)	Subjects with scores below or equal to the 25th percentile were coded as 1 (at risk) and the rest were coded as 0 (no risk).	
PCR	Family	1 (totally disagree) to 5 (totally agree)	Subjects with scores below or equal to the 25th percentile were coded as 1 (at risk) and the rest were coded as 0 (no risk).	
SC	School	1 (totally disagree) to 5 (totally agree)	Subjects with scores below or equal to the 25th percentile were coded as 1 (at risk) and the rest were coded as 0 (no risk).	
TSR	School	1 (totally disagree) to 5 (totally agree)	Subjects with scores below or equal to the 25th percentile were coded as 1 (at risk) and the rest were coded as 0 (no risk).	
PR	Peer	1 (totally disagree) to 5 (totally agree)	Subjects with scores below or equal to the 25th percentile were coded as 1 (at risk) and the rest were coded as 0 (no risk).	
NRF	0	1	2	3	4	5	6	7	8	
NP	37	52	87	49	44	25	14	5	1	
Proportion (%)	11.78	16.56	27.71	15.61	14.01	7.96	4.46	1.59	0.32	
Notes.

FES Family economic status

FLE Father’s level of education

MLE Mother’s level of education

PMR Parental marital relationship

PCR Parent–child relationship

SC School connectedness

TSR Teacher–student relationship

PR Peer relationship

NRF Number of risk factors

NP Number of people

(1) Family economic status: Adopted from the Chinese version of the Family Economic Stress Scale compiled by Wadsworth & Compas (2002) and revised by Wang, Li & Zhang (2010), the scale measures the perception of family economic conditions by children; it consists of four items on a 5-point scale, with higher scores indicating greater family economic hardship. The Cronbach’s α coefficient for the scale in this study was 0.82.

(2) Parental education level: Two items were used to measure the educational level of fathers and mothers (Gerard & Buehler, 2004). A six-point scale was used, ranging from “1 = never attended school” to “6 = graduate student”.

(3) Parental marital relationship: Parental marital relationship was evaluated using the two items cited in the study of Bao et al. (2014) including “Do your father and mother argue?”, which was reverse scored. The Cronbach’s α coefficient was 0.7.

(4) Parent–child relationship: Using the Parent-Child Relationship Subscale of the Chinese version of the Social Relationship Network Questionnaire revised by Hou (1997). The Cronbach’s α coefficient of the scale was 0.92 in this study.

(5) School connectedness: The Chinese version of the School Connection Scale developed by Resnick et al. (1997) and revised by Yu et al. (2011) was used. In this study, the Cronbach’s α coefficient of the questionnaire was 0.86.

(6) Teacher–student relationship: The teacher–student relationship subscale of the Social Relationship Network Questionnaire was used (Hou, 1997). The Cronbach’s α coefficient of the questionnaire was 0.89.

(7) Peer relationship: The peer relationship subscale of the Social Relationship Network Questionnaire was used to measure the level of peer relationship (Hou, 1997). The Cronbach’s α coefficient of the questionnaire was 0.92.

As shown in Table 1, 28.34% of the migrant children experienced four or more ecological risk factors.

Self-education expectations

The question “What academic level do you want to be able to reach?” was used to measure the level of education that the migrant children wanted to achieve in the future (Hong & Ho, 2005). The participants were asked to choose from the following five levels: junior high school, high school (including vocational high school or junior college), university (college or bachelor’s degree), master’s degree, and doctorate (1–5 points).

Internet game addiction

The Internet Gaming Addiction Scale, revised by Chinese scholars based on the scales by Pontes & Griffiths (2015), is widely used in China to measure the level of internet gaming addiction among adolescents and has good reliability and validity (Zeng & Jiang, 2016; Zhang et al., 2023). It contains nine items (e.g., “I feel irritable when I cannot play online games”) on a 5-point Likert scale. The mean scores were calculated for all items, with higher scores indicating a higher tendency to become addicted to Internet gaming. The Cronbach’s α coefficient for the scale in this study was 0.86.

Procedures

The study was approved by the Ethics Committee of Longyan University (LY2024010L). After obtaining informed consent from students, and students’ guardians, group assessments were conducted on a class-by-class basis. The survey was anonymous and the principle of research confidentiality was emphasized. Participants read the instructions carefully and completed the questionnaire as required. The questionnaire took about 40 min to complete.

Data analysis

Descriptive statistics, correlation analyses, and regression analyses were performed using SPSS 24.0, and moderated mediated effects tests were performed using the PROCESS macro.

Results

Common method bias

The data in this study were collected from self-reports of migrant children, and the results may be affected by common methodological biases. Therefore, prior procedural controls such as separating the different questionnaires and setting some questions to be reverse scored were adopted in the design and data collection process. In addition, a post-hoc statistical test for common method bias was conducted using the Harman one-way test (Podsakoff et al., 2003). The results showed that 23 factors had an eigenroot greater than 1, and the first factor explained 19.383% of the variance, which was much less than the critical value of 40%, indicating that the common method bias was not significant.

Correlation analysis

Table 2 presents the mean, standard deviation, and correlation matrix for each variable. There was a significant positive correlation between both cumulative ecological risk and Internet game addiction (r = 0.331, p < 0.001), a significant negative correlation between self-educational expectations and Internet game addiction (r = −0.244, p < 0.001), and a significant negative correlation between cumulative ecological risk and self-educational expectations (r = −0.186, p < 0.01). The results indicate suitability for further mediation effects analyses.

Table 2 Descriptive statistics and correlation matrix for each variable.

Variable	1	2	3	
1. CER	1			
2. SEE	−0.186**	1		
3. IGA	0.331***	−0.244***	1	
M	2.55	3.44	2.42	
SD	1.73	0.99	0.88	
Notes.

CER Cumulative ecological risk

SEE Self-education expectations

IGA Internet gaming addiction

** p < 0.01.

*** p < 0.001.

Relationship between cumulative ecological risk and Internet game addiction: a moderated mediation model test

Regression analysis was used to determine the direct effect of cumulative ecological risk on Internet game addiction. As shown in Table 3, after controlling for grade and gender variables, cumulative ecological risk significantly and positively predicted Internet game addiction (β = 0.148, p < 0.001), indicating that cumulative ecological risk has a direct predictive effect on Internet game addiction, the direct effect is 0.13.

Table 3 Cumulative ecological risk predicting Internet gaming addiction.

	R	R 2	F	β	SE	t	
	0.486	0.236	31.883***				
Gender				−0.602	0.088	−7.074***	
Grade				0.048	0.053	0.908	
CER				0.148	0.025	5.818***	
Notes.

*** p < 0.001.

Table 4 shows that after including the mediating variable of self-educational expectations, cumulative ecological risk significantly and positively predicted Internet game addiction among migrant children (β = 0.134, p < 0.001), and self-educational expectations significantly and negatively predicted Internet game addiction (β = −0.139, p < 0.01). As the direct effect of cumulative ecological risk on Internet game addiction was significant, it can be said that self-educational expectations partially mediate the relationship between cumulative ecological risk and Internet game addiction.

Table 4 Cumulative ecological risk and self-education expectations predict Internet gaming addiction.

Predictive variable	R	R 2	F	β	SE	t	
	0.509	0.259	26.962***				
Gender				−0.602	0.088	−6.869***	
Grade				0.035	0.053	0.672	
CER				0.134	0.026	5.27***	
SEE				−0.139	0.045	−3.09**	
Notes.

** p < 0.01.

*** p < 0.001.

Based on the significance of the mediating effect of self-educational expectations, Model 14 of the PROCESS macro program was used to test the moderating role of gender on the second half of the pathway of the relationship among cumulative ecological risk → self-educational expectations → Internet gaming addiction. As shown in Table 5, the general model fit was significant (F = 28.337, p < 0.001). The interaction term between self-educational expectations and gender positively predicted Internet gaming addiction (β = 0.191, t = 2.31, p < 0.05), suggesting that gender moderates the relationship between self-educational expectations and Internet gaming addiction. The index of moderated mediation was −0.02, The 95% confidence interval was [−0.046, −0.002]. The confidence interval does not contain 0, indicating that the moderated mediation effect is significant.

Table 5 Moderating role of gender.

Predictor variables	R	R 2	F	β	SE	t	95% CI	
	0.518	0.268	28.337***					
CER				0.132	0.025	5.218***	[0.08, 0.18]	
SEE				−0.408	0.133	−3.069**	[−0.67, −0.15]	
Gender				−0.609	0.087	−6.996***	[−0.78, −0.44]	
SEE×Gender				0.191	0.09	2.13*	[0.01, 0.37]	
Notes.

* p < 0.05.

** p < 0.01.

*** p < 0.001.

A simple slope test was used to further explore the moderating effect of gender between self-educational expectations and Internet gaming addiction. The results showed that both males (simple slop = −0.478, t = −3.468, p < 0.01) and females (simple slop = −0.266, t = −4.206, p < 0.001) showed a decrease in online gaming addiction as self-education expectation increased. The decline was more pronounced in boys than in girls (see Fig. 2). It can be seen that self-education expectation has a greater impact on Internet game addiction in males than in females’.

Figure 2 The moderating role of gender.

Discussion

The study, by revealing the impact mechanism of cumulative ecological risk on the Internet gaming addiction of migrant children, deepens the application of the ecological systems theory in the field of digital behavior, providing new evidence for understanding the superimposed effects of multidimensional environmental pressures. The introduction of self-educational expectations as a mediating variable compensates for the insufficiency of traditional studies in exploring internal mechanisms. By verifying the moderating effect of gender, the study emphasizes the differentiated needs of intervention strategies, providing a theoretical basis for gender-specific mental health services. Focusing on the group of migrant children, the study expands the ecological research framework of internet addiction among vulnerable groups, offering practical insights for the formulation of precise policies.

The effect of cumulative ecological risk on migrant children’s Internet gaming addiction

We found that cumulative ecological risk affects online game addiction among migrant adolescents, suggesting that the cumulative ecological risk is a risk factor for the growth of migrant children. The higher the level of cumulative risk in family, school, and peer environments, the more likely migrant children are to develop online game addiction. Hypothesis 1 is valid and supports the conclusion that cumulative ecological risk predicts Internet addiction (Li et al., 2016; Tian et al., 2023). Individuals show many behavioral problems when their environmental conditions are poor (Mason et al., 2019). Under the long-term influence of risk factors, migrant children may gradually develop a stable pattern of negative behavior that cannot be alleviated or subsided in the short term. It may be difficult for migrant children, who experience frequent relocation and environmental changes, to find a sense of belonging and stability in their new environments, to establish effective connections with others, and to develop self-confidence and a sense of control. They may feel frustrated and helpless when facing new challenges and difficulties. In such cases, the Internet provides a safe haven. The virtual nature of online games can provide a space for in-depth communication in which they can vent and release their stress without any worries. It is also possible to escape from reality with the help of online games and seek comfort after being frustrated. The difference between the virtual world and the real world in the game may also lead migrant children to become dependent on Internet games (Tian et al., 2018).

The mediating role of self-educational expectations

The results indicated that self-educational expectations partially mediated the relationship between cumulative ecological risk and Internet game addiction. Cumulative ecological risk adversely affected self-educational expectations, while self-educational expectations negatively predicted online game addiction. Thus, research hypothesis 2 was verified.

The cumulative ecological risk reflects the serious lack of current supportive ecological resource environments for migrant adolescents, and the higher their perceived level of cumulative ecological risk, the lower their expectations for the achievement of future academic goals. Mobility disrupts children’s original school, family, and peer ecosystems, resulting in unstructured and disorganized systems that affect the social interactions between migrant children and members of the ecological microsystem. Poor ecosystems undermine adolescents’ self-identity, fail to effectively stimulate their growth initiatives, make it difficult for them to adapt to the current society, and impact their confidence in future development, in turn leading to a reduction in positive future-oriented beliefs. Cumulative risk factors in the environment have an impact on self-efficacy and future expectations. When the cumulative ecological risk is low, support from family, teachers, and peers deepens students’ understanding of learning goals, and their potential for self-education is stimulated, and this can significantly increase students’ future expectations. Positive academic expectations may have a protective effect against risky behavior (Lee et al., 2015).

According to the expectancy effect, students with high self-expectations have clear academic goals, are able to organize their lives more rationally, and have high levels of self-control. They hope to achieve their self-education expectations through continuous efforts and by devoting themselves to their studies; they take the initiative to reduce the interference of other events (e.g., recreational and leisure games). They are more positive and optimistic about their future academic expectations, can objectively know themselves and adjust their self-status, and have a psychological tendency to pursue behaviors relating to delayed gratification (e.g., healthy behaviors such as physical exercise and reasonable time arrangement). On the contrary, migrant children have negative and pessimistic expectations of their own academic achievements, often lack systematic and reasonable planning of time, tend to be keen on instant gratification, and are likely to be attracted by games on the Internet. At the same time, due to the lack of external supervision and low self-control, they gradually become addicted to Internet games (Jin, Ji & Li, 2022).

The moderating role of gender

The results indicate that self-educational expectations have a greater impact on Internet gaming addiction among boys than among girls, and the moderated mediation model is supported. This result is also in line with the results of previous studies on gender differences in Internet addiction (Hyun et al., 2015; Toker & Baturay, 2016).

On the one hand, junior high school is a critical period for the development of self-awareness and gender differences. Self-awareness and self-control development levels are higher among adolescent females than among their male counterparts (Zhou & Zhang, 2012). Girls are generally more mentally mature than boys of the same age group, have a more comprehensive understanding of themselves, and are eager to change their unfavorable situation through learning, and therefore have clearer goals for the future.

On the other hand, from the perspective of evolutionary psychology, males have faced more stressors in society since ancient times. In the face of future academic planning, males have higher negative feelings towards themselves than females, and males are more likely to lack learning goals and motivation. When anxiety arises in real interactions that cannot be resolved, males are more likely to turn to the Internet for entertainment. Online games present a virtual competitive environment that can demonstrate the participants’ skills and strengths and compensate for the sense of loss in reality. In addition, online games also provide many achievements and rewards, which motivate males to pursue success and progress, and in the process, male students with low self-control easily become addicted to Internet games (Liang et al., 2016). Therefore, more attention should be paid to migrant males and their future expectations should be correctly guided.

Implications

Migrant adolescents are subject to multiple ecological risks in their families and schools, and measures should be taken to reduce the number of high-risk factors in the ecological sub-system, and a highly structured and supportive ecological background is conducive to preventing and improving the problem of online game addiction among migrant adolescents. In terms of family risk, although parents of migrant families are busy with their livelihoods and generally have a low level of education, they do not have enough time and energy to accompany their children. However, they should not neglect to create a harmonious environment for their children to grow up in, provide sufficient emotional support, and communicate with and enlighten their children as a means of mitigating the negative impact of cumulative family risks. Parents should pay attention to cultivating young people’s reasonable perception of the consequences of Internet use, so as to make them aware of the potential adverse consequences of Internet use. In terms of school and peer risks, schools should organise a wide variety of activities to enhance students’ connection with the school and the sense of belonging and community among migrant youth. Teachers should guide local children to reduce the “stigmatisation” of migrant youth and promote peer support among students. Boys’ self-educational expectations have a greater impact on Internet gaming addiction than those of girls, which inspires us to actively guide boys in order to help them develop reasonable educational goal orientations.

Limitations and prospects

Firstly, the present study is a cross-sectional study and cannot infer causal relationships between variables. Future research could use follow-up studies and intervention experiments to better test the mediation model developed in this study. Second, this study focuses on the internal mobile children in China. Since there may be differences in the ecological risks faced by international and internal migrants, future research can further explore the international migrant population. Thirdly, this study focuses on the mobile children in Fujian Province; the selection of this specific sample may limit the generalizability of the research results to a broader group of mobile adolescents. In future studies, it will be necessary to expand the sample size to more comprehensively assess the universal applicability of the research. Finally, although the ecological risk factors selected in this study are typical and representative, they do not fully summarise all potential risk factors in adolescent development, and future research can further refine the cumulative risk factors to better test the findings of this study.

Conclusions

Our study sheds light on the intricate relationship between cumulative ecological risk, self-educational expectations, and Internet game addiction among migrant students, with a particular focus on the moderating role of gender. The findings emphasize the significance of considering gender differences in addressing the issue of Internet addiction, especially among migrant students who may face heightened ecological risks. This work contributes to the growing body of research on adolescent Internet addiction and underscores the need for a comprehensive and multifaceted approach to addressing this pressing public health concern.

Supplemental Information

Supplemental Information 1 Data of migrant children

Supplemental Information 2 Codebook

Supplemental Information 3 Questionnaire

The authors sincerely thank all the students who have participated in this study.

Additional Information and Declarations

Competing Interests

Author Contributions

Human Ethics

Data Availability

The authors declare there are no competing interests.

Zhengzheng Lin conceived and designed the experiments, performed the experiments, analyzed the data, prepared figures and/or tables, authored or reviewed drafts of the article, and approved the final draft.

Ying Zha analyzed the data, authored or reviewed drafts of the article, and approved the final draft.

The following information was supplied relating to ethical approvals (i.e., approving body and any reference numbers):

The study was conducted in accordance with the Declaration of Helsinki, and approved by the Ethics Committee of Longyan University (protocol code LY2024010L).

The following information was supplied regarding data availability:

The raw measurements are available in the Supplementary File.

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
