# Peer review of "The effect of cumulative ecological risk on migrant children’s Internet game addiction: a moderated mediation model"

_PeerJ, doi:10.7717/peerj.19787_

## Round 0.1 · original submission · Major Revisions

· Academic Editor

Major Revisions

Please address all the comments of the reviewers

Reviewer 1 ·

Basic reporting

1) Line 45: There is insufficient evidence to support the claim that gaming is the most addictive activity on the internet. In fact, a meta-analysis found a pooled prevalence rate of 3.3% for internet gaming disorder (Kim et al., 2022). In contrast, the pooled prevalence rate for social media addiction was 5% (Cheng et al., 2021).
2) Line 71: The sentence should be: “On the family side, with inadequate parental support…”
3) Lines 71 to 79 and 153 to 166: A distinction should be made general addictions (e.g., internet addiction and smartphone addiction) and specific addictions (e.g., internet gaming disorder) since these are related but distinct constructs (Griffiths, 2014; Király et al., 2014; Sigerson et al., 2017; Tan & Chew, 2024). Only research related to internet gaming disorder should be included in the introduction.
4) Since gaming addiction was assessed using the DSM-5 Internet Gaming Disorder criteria, the term ‘internet gaming disorder’ should be use and the nine criteria should be described.
5) Might be worth standardizing and using the term “migrant children” throughout the manuscript instead of “children on the move” (lines 58 to 59) and “mobile children” (line 121).
6) Some sentences should be rephrased for grammar and clarity (e.g., lines 40 to 44, 63 to 64, etc.).

Experimental design

1) The term “migrant children” should be defined in the study. Specifically, Unicef defined migrant children as “…people living in a country outside their country of birth…” (https://data.unicef.org/topic/child-migration-and-displacement/migration/). In this study, it appears that the migrant children are also from China but from a different province. If so, the amount of ecological risk, while significant, might be lower than if they were from a different country.
2) Line 210: The previous studies in question should be cited and referenced.
3) Line 213 to 216: The rationale for dichotomizing each risk factor instead of using them as continuous variables could be provided.
4) Were the instruments administered in Chinese? If so, was the back translation procedure used for all instruments?
5) It is unclear if the instruments were suitable for or validated among children. First, the total number of items and average duration of the study should be included with a comment on their implications for participants’ fatigue. Second, the participants might not understand some of the items. For example, Item 1 of the IGDS reads “Do you feel preoccupied with your gaming behaviour? (Some examples: Do you think about previous gaming activity or anticipate the next gaming session? Do you think gaming has become the dominant activity in your daily life?)”. This appears to be beyond the comprehension level of 13-year-old children.

Validity of the findings

1) Line 289: Perhaps the author could clarify how ‘cumulative ecological risk’ was scored.
2) Line 292: Table 4 does not show the results of a mediation. Instead, it shows the results of either simple linear regression or multiple regression. Furthermore, the title and introduction set the stage for a moderated mediation model, so a standalone mediation analysis is not required.
3) The reporting of the PROCESS analysis could be improved by providing more details (e.g., statistics associated with the general model, variance explained, and direct and indirect effect).
4) The statistical diagram of the model, along with the statistics, should be provided.

Additional comments

References
Cheng, C., Lau, Y., Chan, L., & Luk, J. W. (2021). Prevalence of social media addiction across 32 nations: Meta-analysis with subgroup analysis of classification schemes and cultural values. Addictive Behaviors, 117, 106845. https://doi.org/10.1016/j.addbeh.2021.106845
Griffiths, M. D. (2014). Internet addiction disorder and internet gaming disorder are not the same. Journal of Addiction Research & Therapy, 05(04). https://doi.org/10.4172/2155-6105.1000e124
Kim, H. S., Son, G., Roh, E.-B., Ahn, W.-Y., Kim, J., Shin, S.-H., Chey, J., & Choi, K.-H. (2022). Prevalence of gaming disorder: A meta-analysis. Addictive Behaviors, 126, 107183. https://doi.org/10.1016/j.addbeh.2021.107183
Király, O., Griffiths, M. D., Urbán, R., Farkas, J., Kökönyei, G., Elekes, Z., Tamás, D., & Demetrovics, Z. (2014). Problematic internet use and problematic online gaming are not the same: Findings from a large nationally representative adolescent sample. Cyberpsychology, Behavior, and Social Networking, 17(12), 749–754. https://doi.org/10.1089/cyber.2014.0475
Sigerson, L., Li, A. Y.-L., Cheung, M. W.-L., & Cheng, C. (2017). Examining common information technology addictions and their relationships with non-technology-related addictions. Computers in Human Behavior, 75, 520–526. https://doi.org/10.1016/j.chb.2017.05.041
Tan, C. S. Y., & Chew, P. K. H. (2024). General addiction versus specific addiction: Which is associated with a higher risk of negative consequences. Current Psychology. Advance online publication. https://doi.org/10.1007/s12144-024-07015-z

Reviewer 2 ·

Basic reporting

I only have one minor comment regarding the data base. To better understand the results related with the variable gender, how was gender coded? I think I inferred male was coded as 1 and female as 0. This detail may help interpret the sign of the betas obtained with the variable "gender".

Experimental design

No comment

Validity of the findings

No comment

Additional comments

The Table 3 is a bit confusing to me. It gives the impression that IGA is a predictive variable. Should not be CER instad of IGA? The rest of the tables are clear to me.

Reviewer 3 ·

Basic reporting

How is the role of moderation and mediation in preventing internet gaming disorder in migrant children, there is no explanation yet

Experimental design

No Comment

Validity of the findings

In the research results, can you explain the order of the strongest factors first and also the main cause in sequence of the four variables (internet gaming disorder, cumulative ecological risk, independent education expectations and gender.

In the discussion, it is necessary to add how researchers contribute benefits to the development of science in your field.

Additional comments

Use clear, unambiguous, professional English throughout, with proofreading possible.

·

Basic reporting

Title and Abstract: The title is clear and effectively reflects the study’s focus on cumulative ecological risk and Internet game addiction. The abstract is concise but could be more specific in detailing the methodology and sample size to better inform the reader. While the objectives and key findings are stated, a clearer description of the mediating and moderating variables would improve clarity.

Introduction: The introduction establishes the context of Internet gaming addiction among migrant children and discusses the significance of ecological risks. The rationale for the study is generally strong, citing relevant prior research. However, more detail on the gaps in existing literature that this study seeks to address would enhance the reader’s understanding of the study's contribution. The hypothesis could be more explicitly stated.

Experimental design

Sample: The study used a convenience sampling method from migrant children in grades 7-9, with a high response rate of 98.86%. This is a strong point, ensuring a representative sample. However, further details about the demographic diversity of the participants (e.g., geographic distribution, socioeconomic status) would provide more context.

Measures: The choice of measures appears appropriate for examining cumulative ecological risk, self-educational expectations, and Internet game addiction. The use of various well-established scales (e.g., Family Economic Stress Scale, School Connection Scale) strengthens the study's foundation. However, more information on how the cumulative ecological risk index was calculated would be useful for transparency.

Data Collection: The data were collected through self-reported questionnaires in a classroom setting. While this is standard practice, self-reports can introduce bias, especially in sensitive topics like Internet addiction. Although the study addresses this issue with common method bias tests, further details on the bias control mechanisms would be helpful. The use of reverse-scored questions and separated questionnaires for different measures was a good strategy.

Validity of the findings

Internal Validity: The study design is sound, and statistical analyses (descriptive statistics, correlation analysis, and regression models) are appropriate for testing the hypotheses. The moderation and mediation models help clarify the relationships between variables. However, since the study is cross-sectional, causality cannot be conclusively established. The authors acknowledge this limitation, but future longitudinal or experimental studies are necessary to better establish causal links.

External Validity: The study’s focus on migrant children in a specific region (Fujian Province) may limit generalizability to other populations. The use of a convenience sample also limits the ability to generalize the findings to a broader adolescent population. A more diverse sample could strengthen the external validity.

Construct Validity: The scales used to measure cumulative ecological risk, self-educational expectations, and Internet game addiction appear reliable (high Cronbach’s alpha coefficients). However, the cumulative ecological risk index could benefit from further explanation regarding how the individual risk factors were combined and why those particular factors were chosen.

Statistical Validity: The statistical analyses conducted (e.g., regression analysis, mediation, moderation) are appropriate for the data type. The effect sizes for regression coefficients would strengthen the interpretation of results. The Harman’s one-way test for common method bias indicates that method bias is not a significant issue.

Additional comments

Strengths:

The research addresses a timely and important issue concerning Internet addiction among migrant children, an often under-researched group.
The study design is robust, using validated measures for the key variables and demonstrating the relationships between ecological risk factors, self-educational expectations, and Internet addiction.
The inclusion of gender as a moderating factor is insightful, adding depth to the findings and providing a nuanced understanding of the phenomenon.
Suggestions for Improvement:

The introduction could be more explicit in stating the research gaps and hypothesis to guide the reader more clearly.
While the methodology is detailed, a clearer explanation of the calculation of the cumulative ecological risk index would add transparency.
The authors should discuss potential alternative explanations for the findings (e.g., other environmental or individual factors not considered in the study).
The authors should address the study's cross-sectional design more thoroughly, considering its limitations in establishing causality.
The discussion could benefit from a more critical examination of limitations, particularly regarding the sample's representativeness and the self-report data collection method.
Implications:

The study provides valuable insights into the ecological risk factors influencing Internet gaming addiction, particularly in migrant adolescents. The findings highlight the importance of addressing environmental factors such as family, school, and peer relationships to prevent or reduce Internet addiction.
Practical recommendations for interventions are relevant, but more detail on how these recommendations could be implemented in real-world settings (e.g., schools and families) would strengthen the practical application of the findings.

---

## Round 0.2 · Minor Revisions

· Academic Editor

Minor Revisions

Thank you for your revision. Reviewer 2 has a minor comment to address.

Reviewer 2 ·

Basic reporting

I appreciate the many efforts the authors made to improve the manuscript. I believe the manuscript improved considerably. I only have one very minor comment: in this new version of the manuscript, the term "Internet gaming addiction" appears many times; perhaps the authors would want to use the acronym IGD to make the text more readable.

Experimental design

No comment

Validity of the findings

No comment

---

## Round 0.3 · accepted · Accept

· Academic Editor

Accept

The reviewer notes, and I concur, that you have addressed all the reviewer comments, and that the manuscript is now ready for publication.

Reviewer 2 ·

Basic reporting

No comment

Experimental design

No comment

Validity of the findings

No comment

Additional comments

Thanks to the authors for their attention. I have no further comments regarding the manuscript.